# Echinochrome A Prevents Diabetic Nephropathy by Inhibiting the PKC-Iota Pathway and Enhancing Renal Mitochondrial Function in db/db Mice

**DOI:** 10.3390/md21040222

**Published:** 2023-03-30

**Authors:** Trong Kha Pham, To Hoai T. Nguyen, Hyeong Rok Yun, Elena A. Vasileva, Natalia P. Mishchenko, Sergey A. Fedoreyev, Valentin A. Stonik, Thu Thi Vu, Huy Quang Nguyen, Sung Woo Cho, Hyoung Kyu Kim, Jin Han

**Affiliations:** 1Department of Physiology, Cardiovascular and Metabolic Disease Center, Smart Marine Therapeutic Center, College of Medicine, Inje University, Busan 47392, Republic of Korea; 2Faculty of Biology, University of Science, Vietnam National University, Hanoi 10000, Vietnam; 3G.B. Elyakov Pacific Institute of Bioorganic Chemistry, Far-Eastern Branch of the Russian Academy of Science, 690022 Vladivostok, Russia; 4Division of Cardiology, Department of Internal Medicine, Ilsan Paik Hospital, Cardiac & Vascular Center, College of Medicine, Inje University, Goyang 10380, Republic of Korea

**Keywords:** echinochrome A, diabetic nephropathy, protein kinase C, renal fibrosis, oxidative stress

## Abstract

Echinochrome A (EchA) is a natural bioproduct extracted from sea urchins, and is an active component of the clinical drug, Histochrome^®^. EchA has antioxidant, anti-inflammatory, and antimicrobial effects. However, its effects on diabetic nephropathy (DN) remain poorly understood. In the present study, seven-week-old diabetic and obese db/db mice were injected with Histochrome (0.3 mL/kg/day; EchA equivalent of 3 mg/kg/day) intraperitoneally for 12 weeks, while db/db control mice and wild-type (WT) mice received an equal amount of sterile 0.9% saline. EchA improved glucose tolerance and reduced blood urea nitrogen (BUN) and serum creatinine levels but did not affect body weight. In addition, EchA decreased renal malondialdehyde (MDA) and lipid hydroperoxide levels, and increased ATP production. Histologically, EchA treatment ameliorated renal fibrosis. Mechanistically, EchA suppressed oxidative stress and fibrosis by inhibiting protein kinase C-iota (PKCι)/p38 mitogen-activated protein kinase (MAPK), downregulating p53 and c-Jun phosphorylation, attenuating NADPH oxidase 4 (NOX4), and transforming growth factor-beta 1 (TGFβ1) signaling. Moreover, EchA enhanced AMPK phosphorylation and nuclear factor erythroid-2-related factor 2 (NRF2)/heme oxygenase 1 (HO-1) signaling, improving mitochondrial function and antioxidant activity. Collectively, these findings demonstrate that EchA prevents DN by inhibiting PKCι/p38 MAPK and upregulating the AMPKα/NRF2/HO-1 signaling pathways in db/db mice, and may provide a therapeutic option for DN.

## 1. Introduction

Diabetes mellitus is one of the most common chronic diseases and is a serious public health problem and cause of death worldwide [1]. Its global prevalence in 20- to 79-year-olds was estimated to be 537 million people in 2021 and is expected to increase to 784 million in 2045 [2]. Type 2 diabetes mellitus accounts for ≥90% of patients with diabetes [2]. Uncontrolled diabetes can lead to various complications, such as diabetic nephropathy (DN), a major cause of end-stage renal disease (ESRD), which occurs in approximately 40% of patients with diabetes [2,3]. In addition, diabetic renal disease is a risk factor for cardiovascular diseases, resulting in mortality and morbidity [4].

DN progression occurs in several stages. Early changes in the kidneys of patients with diabetes involve marked glomerular hyperfiltration and hypertrophy, followed by impairment of the glomerular filtration barrier, an increase in urinary albumin excretion, mesangial matrix accumulation, hypertrophy, nodular glomerulosclerosis, and interstitial fibrosis, which is the final step toward ESRD and kidney failure [5,6,7,8]. Oxidative stress plays a key role in the pathogenesis of DN [9,10]. Previous reports have demonstrated that excessive hyperglycemia-induced reactive oxygen species (ROS) production triggers renal fibrosis and inflammation. This leads to significant tissue damage by promoting malondialdehyde (MDA), DNA damage, protein modification, and mitochondrial dysfunction, resulting in a decrease in ATP production [11,12,13]. In addition, among the many enzyme systems involved in ROS generation in the kidney, NADPH oxidases (NOXs) appear to be the most important [14,15]. Moreover, ROS can promote renal sclerosis, fibrosis, and oxidative stress by activating transforming growth factor-beta (TGF-β) and NOX signaling, accompanied by the upregulated activation of protein kinase C (PKC) and the subsequent activation of mitogen-activated protein kinase (MAPK) [5,16,17]. PKC belongs to a family of serine/threonine-specific protein kinases with various isoforms that induce glomerular damage and oxidize the structure of several molecules, including DNA, carbohydrates, lipids, and proteins [18,19,20,21,22].

Although there are various therapies for glucose control to retard the progression of the complications of diabetes, including DN, many patients with diabetes remain at high risk for developing and progressing DN. Therefore, there is an urgent need for more effective therapeutic strategies to prevent DN development and progression, in which molecules extracted from natural sources remain to be investigated.

Echinochrome A (EchA, 7-ethyl-2,3,5,6,8-pentahydroxy-1,4-naphthoquinone), a bioactive dark-red pigment extracted from the shells and spines of sea urchins, is registered as a medicinal product and approved for medicinal use in Russia (reg. no. P N002363/01) and is the active substance of the drug Histochrome^®^. Histochrome^®^ has been approved for medicinal use in Russia against various diseases, including cardiac ischemia, myocardial infarction, middle cerebral artery occlusion, hepatopathy, and several ophthalmic diseases [23,24,25,26,27,28,29]. Through a broad investigation of its biological effects, the antioxidant, anti-inflammatory, and antibacterial abilities of EchA have been reported [24,27,30,31]. A study of Huixing Cui et al. recently showed that EchA reduced blood pressure and kidney abnormalities in a rat model of preeclampsia by modulating inflammation and apoptosis [32]. However, the specific target and mechanism of action of EchA remain unclear. Moreover, the potential renal protective effects of EchA have not yet been evaluated in a diabetic db/db mouse model of DN. We previously discovered that EchA could bind directly to PKC-iota (PKCι) and inhibit its activity [33]. Therefore, the present study was conducted to elucidate the signaling mechanism of EchA in preventing DN in db/db mice. We found that EchA prevents DN in db/db mice by inhibiting PKCι activation, downregulating renal fibrosis and oxidative stress-related signals, and upregulating the AMPKα/NRF2/HO-1 pathway, thereby contributing to improved renal mitochondrial function. The obtained data suggest the potential reno-protective mechanism of EchA in DN.

## 2. Results

### 2.1. EchA Reduced Blood Urea Nitrogen and Creatinine Levels and Improved Blood Glucose Tolerance in db/db Mice

The effects of EchA on body weight, blood glucose, blood urea nitrogen (BUN), creatinine, and insulin levels in db/db mice are presented in Figure 1. The results showed that there was no significant difference in body weight between db/db and db/db + EchA mice (Figure 1B). Seven-week-old db/db mice exhibited higher blood glucose levels than same-aged WT mice. After 12 weeks of treatment, EchA improved glucose tolerance, although it did not affect fasting blood glucose levels (Figure 1C,D). An analysis of the blood samples of the mice revealed that the serum levels of BUN and creatinine were lower in the WT and db/db + EchA groups than in the db/db group (*p* < 0.05 and *p* < 0.01, respectively) (Figure 1E,F). However, insulin levels were not significantly different among the groups (Figure 1G). The results indicate that EchA improves kidney function in diabetic mice.

### 2.2. EchA Treatment Attenuated Renal Hypertrophy and Fibrosis in db/db Mice

The kidneys of db/db mice were clearly larger, and the kidney-weight-to-tibia-length ratio was higher than that of WT mice, but it was decreased in db/db + EchA mice (*p* < 0.01 vs. db/db mice) (Figure 2A,B). In addition, hematoxylin and eosin (HE) staining of the kidneys showed that the glomerular capillaries diminished, glomerular diameters were enlarged, and the ratios of the glomerular diameter to renal capsule volume were the highest in the db/db mice group. However, these parameters were attenuated in the kidneys of EchA-treated db/db mice (*p* < 0.05 vs. non-treated db/db mice) (Figure 2C,D). Moreover, Masson’s trichrome (MT) staining showed that renal interstitial fibrosis and accumulation of collagen fibers were increased in the kidneys of db/db mice compared to those in WT mice. However, these were attenuated after 12 weeks of EchA treatment in db/db + EchA mice (*p* < 0.05 vs. db/db mice) (Figure 2C,E). To further verify these renal changes, we conducted Western blot analysis to determine the protein expression of TGF-β1, Col3, α-SMA, and SMAD2 phosphorylation. The results showed an increase in the expression of these proteins in the kidneys of db/db mice. TGF-β1 and Col3 levels were attenuated (*p* < 0.05 vs. db/db mice) (Figure 2F–H), and the expression of α-SMA and SMAD2 phosphorylation was also slightly reduced after EchA treatment (Figure 2F,I,J). These results demonstrate that EchA treatment inhibited renal hypertrophy and fibrosis in db/db mice.

### 2.3. EchA Ameliorated Renal Oxidative Stress in Diabetic Mice

Excessive oxidative stress is a significant inducer of DN in response to high glucose levels. To determine the effect of EchA on oxidative stress in diabetic mice, we measured MDA and lipid hydroperoxide levels and the protein expression levels of SOD1, SOD2, NOX2, and NOX4 in renal tissues. The oxidative stress parameters are shown in Figure 3. We found that MDA and lipid hydroperoxide levels were increased in the renal tissue of db/db mice compared with those in the WT group, while EchA reduced these levels (*p* < 0.05 vs. db/db mice) (Figure 3A,B). NOX is the primary source of ROS in DN [34]. As expected, a significant increase in NOX4 expression was observed in diabetic kidneys compared with the WT group. EchA treatment reversed NOX4 expression but not NOX2 expression in the db/db + EchA group (Figure 3C–E). In addition, the results showed that the expression levels of the antioxidant enzymes, SOD1 and SOD2, were decreased in the db/db group relative to the WT group. In EchA-treated diabetic mice, SOD1 expression was higher than that in untreated mice (*p* < 0.05 vs. db/db mice), while SOD2 expression was not visibly changed (Figure 3C,F,G). Taking these results together, EchA can alleviate excessive oxidative stress by attenuating the levels of oxidation products and elevating the levels of antioxidant enzymes in the renal tissues of diabetic mice.

### 2.4. EchA Activated AMPK Phosphorylation and the NRF2/HO-1 Pathway to Improve Mitochondrial Function and Increase ATP Production in Diabetic Kidneys

The AMP-activated protein kinase (AMPK) system serves as a sensor and protector of cellular energy needs by inhibiting energy consumption and stimulating energy production [35]. To assess the effects of EchA on ATP production and AMPK expression in the kidneys of diabetic mice, we determined the levels of ATP and AMPK expression and phosphorylation. We found that ATP production was decreased in the renal tissue of db/db mice compared to that in the WT group (*p* < 0.01 vs. WT mice), while EchA increased ATP levels in the db/db + EchA group (*p* < 0.05 vs. db/db mice) (Figure 4A). Corresponding to this result, activated AMPK phosphorylation was also elevated in EchA-treated db/db mice compared to that in the untreated group (*p* < 0.05 vs. non-treated db/db mice) (Figure 4B,C). In addition to the role of AMPK, the PGC-1α/NRF2/HO-1 signaling pathway also plays an important role in mitochondrial health and cellular energy metabolism by promoting mitochondrial biogenesis. This study showed that EchA might improve the AMPK/NRF2/HO-1 signaling pathway in diabetic kidneys. Indeed, Western blotting revealed that PGC-1α, NRF2, and HO-1 levels significantly decreased in the renal tissues of db/db mice compared with the WT group. However, PGC-1α expression tended to increase in the EchA-treated group compared to that in the untreated group. Moreover, there was no difference in the expression of PGC-1α between the db/db + EchA and WT groups, while there was a difference in the expression of this protein between the db/db and WT groups (*p* < 0.01 db/db vs. WT mice) (Figure 4B,D). In addition, EchA elevated the protein expression of both NRF2 and HO-1, *p* < 0.05 (Figure 4B,E,F).

### 2.5. EchA Inhibited PKCι/p38MAPK Activation and Downregulated Both p53 and c-Jun Phosphorylation in Diabetic Kidneys

To further explore the potential molecular mechanism of the protective effect of EchA in diabetic nephropathy in vivo, the activity of the PKCι/p38 MAPK pathway and its downstream targets, such as p53 and c-Jun, was examined. There is evidence that the PKC family plays an important role in DN pathogenesis among various signaling kinases [22]. In addition, our previous study showed that EchA could bind directly to PKCι and inhibit its activity [33]. Therefore, we investigated whether EchA could affect PKCι activity in diabetic kidneys. As expected, we found an apparent upregulation of PKCι levels in diabetic kidneys compared to that in the WT group, and EchA treatment reversed this trend in PKCι expression (Figure 5A–C). Additionally, it has been reported that p38 MAPK is a downstream target of PKC in diabetic conditions. The results showed that p38 MAPK phosphorylation increased in the renal tissues of db/db mice, but it was reduced by EchA treatment (*p* < 0.05 vs. db/db mice) (Figure 5A,D). Moreover, recent studies have suggested that p53 may have divergent roles in regulating fibrosis in different animal models [36,37,38] and that c-Jun, a direct downstream target of MAPK signaling, plays a role in the development of diabetes-induced fibrosis, oxidative stress, and apoptosis [39]. We observed an increase in both p53 and c-Jun phosphorylation at serine 15 and 63, respectively, in the kidneys of db/db mice compared with the WT group. However, they were attenuated by EchA treatment (*p* < 0.05 vs. db/db mice) (Figure 5E,F,G).

## 3. Discussion

DN is one of the most serious complications of diabetes, occurring in approximately half of all patients with type 2 diabetes and one-third of all patients with type 1 diabetes. It continues to develop and cause ESRD worldwide, despite many improvements in diabetes care over the past two decades [6,40,41,42]. Unfortunately, there are no drugs available that can effectively treat DN. Thus, there is an urgent need for more effective therapeutic strategies to prevent DN development and progression. A growing number of alternative therapeutic products, including molecules from marine organisms, have been derived from natural sources. EchA, a dark-red pigment found in many sea urchin species, is an active component of Histochrome^®^, which is used for treating various diseases, including cardiovascular, ophthalmic, inflammatory, and metabolic diseases [23,24,25,28,43]. It has been reported that EchA showed antioxidant, anti-inflammation, and antiviral benefits. However, the specific targets of EchA and its mechanism of action in various diseases remain unclear. Additionally, the potential renal protective effects of EchA have not yet been evaluated in a diabetic db/db mouse model of DN. In the present study, we used leptin-receptor-deficient db/db mice, a mouse model of type 2 diabetes, to investigate the effect of EchA on DN progression and gain more insight into the mechanisms underlying the beneficial action of EchA in vivo. We demonstrated that EchA improved glucose tolerance and reduced BUN and serum creatinine levels but did not affect body weight. In addition, EchA decreased renal MDA and lipid hydroperoxide levels and increased ATP production. The key findings of this study are that treatment with EchA ameliorated renal fibrosis and oxidative stress in db/db mice and enhanced mitochondrial function in diabetic kidneys by inhibiting PKCι/p38 MAPK and activating the AMPK/NRF2/HO-1 pathway. This study provides a novel therapeutic strategy for DN.

Among the pathological changes in DN, renal fibrosis is a common metabolic alteration in the late stage of DN [42,44]. Several studies have shown the fibrosis-reducing effect of EchA in several different pathological models [45,46,47]. The TGF-β signaling pathway is known to play a critical role in fibrogenesis, especially in renal fibrosis in DN. In the present study, we showed that EchA ameliorated renal fibrosis and decreased the expression of extracellular matrix proteins, such as collagen III and αSMA, in diabetic kidneys. Moreover, EchA treatment inactivated PKCι/p38 MAPK signaling and suppressed TGF-β1/Smad2 signaling.

In addition, oxidative stress, which plays a crucial role in the pathogenesis of chronic diseases, is characterized as a state in which cell damage due to excessive ROS production in vivo exceeds the natural antioxidant properties of cells, which can damage proteins, lipids, and DNA. Previous studies have confirmed that diabetes can cause elevated ROS levels in vivo [48,49]. One of the sources of ROS under diabetic conditions is the increase in NADPH oxidase activation. Among the different NADPH oxidase isoforms, NOX4 was found to be the main enzyme contributing to increased oxidative stress in podocytes, as genetic ablation or the pharmacological inhibition of NOX4 activity attenuated DN in a rodent model of diabetes [50]. Here, we found that MDA and lipid hydroperoxide levels and NOX4 expression were elevated in the renal tissues of db/db mice, while they were downregulated by EchA treatment. In addition, the expression of the antioxidant enzyme SOD1 was reduced in the renal tissues of db/db mice compared with WT mice, but it was increased by EchA treatment. We explored the potential underlying mechanism and found that EchA may inhibit PKCι signaling and promote the expression of the NRF2/HO-1 pathway to suppress oxidative stress in diabetic kidneys.

AMPK and PGC-1α are key regulators of mitochondrial health, which have been shown to play crucial roles in renal resilience against disease [35]. Energy sensing by AMPK is especially relevant in renal cells because it is strongly dependent on the regulation of energy metabolism for tubular transport. Studies have shown the essential role of AMPKα dysregulation in obesity- and diabetes-associated kidney diseases [35,51,52]. In addition, there is evidence demonstrating the mitochondrial protection effect exerted by AMPK, particularly by activating PGC-1α [53,54]. Moreover, the PGC-1α/NRFs/HO-1 signaling pathway plays a vital role in mitochondrial function. In the present study, EchA enhanced mitochondrial function in db/db mice by activating AMPK phosphorylation and, subsequently, improving the PGC-1α/NRFs/HO-1 signaling pathway, resulting in increased ATP production. Other studies have shown that the upregulation of NRF2/HO-1 signaling also prevents oxidative stress by reducing lipid peroxidation and NOX4 expression [55,56]. Therefore, EchA treatment improved renal mitochondrial function, attenuating oxidative stress in diabetic kidneys.

Next, the potential signaling pathways underlying the effects of EchA on DN were investigated. Among signaling kinases, protein kinase C (PKC) seems to play a crucial role in the pathogenesis of DN [22]. The PKC family is involved in various signal transduction pathways associated with the cell cycle, cell proliferation, differentiation, and apoptosis [57]. However, the role of PKCs in DN has not been fully elucidated. Currently, the PKC family includes 13 isoforms, most of which are expressed in kidney cells [57]. Atypical isoforms of PKCs, including PKCι, are highly expressed in podocytes. However, the role of atypical PKC isoforms in DN remains unclear [57]. Our previous study confirmed that EchA could directly bind to PKCι and inhibit its activity [33]. In the present study, we found an apparent upregulation of PKCι expression in the kidneys of db/db mice compared to that in WT mice, but EchA treatment reversed this trend. In addition, various studies have shown that activated PKC participates in signal transduction pathways by activating several downstream proteins, including the MAPK cascade, which upregulates fibrotic growth factor expression by activating p53 and/or c-Jun [37,38,58,59,60,61]. Here, we found that an increase in PKCι expression upregulated p38 MAPK phosphorylation, which activated both p53 and c-Jun phosphorylation, resulting in the elevated expression of the TGF-β1 signaling pathway, causing renal fibrosis in db/db mice; these signals were significantly decreased by EchA treatment. Moreover, PKC activates NOX4 expression [62]. According to an investigation of Jiao Qin et al., the PKC/NOX pathway is activated in DN [19]. In this study, elevated-PKCι expression may upregulate NOX4, which promotes oxidative stress in the kidneys of db/db mice; treatment with EchA substantially reversed this trend.

Based on the above results, we suggest a schematic diagram depicting the potential reno-protective mechanism of EchA in DN (Figure 6). EchA prevents DN by inhibiting the PKCι/p38 MAPK pathway and enhancing the AMPKα/NRF2/HO-1 pathway in db/db mice. Furthermore, the mechanism of EchA in type 2 diabetes mouse models with distinct genetic backgrounds needs to be explored.

Although, our present study has successfully demonstrated the therapeutic effect of EchA on diabetic nephropathy, along with revealing the underlying molecular mechanism of its therapeutic action. However, we acknowledge that our study has some limitations, including the absence of comparative experiments between multi-dose and positive control drugs. Therefore, further studies will be conducted to evaluate the potential of EchA as a treatment drug for diabetic nephropathy and to compare its effectiveness with other drugs. We look forward to conducting these studies in the near future to better understand the clinical potential of EchA for treating diabetic nephropathy.

## 4. Materials and Methods

### 4.1. Preparation of Echinochrome A

Echinochrome A (7-ethyl-2,3,5,8,8-pentahydroxy-1,4-naphthoquinone) was isolated from the sea urchin *Scaphechinus mirabilis* using a previously described method [63]. The chromato-mass-spectrometric parameters and spectral data of the obtained echinochrome A sample were consistent with those previously published. The purity of the EchA sample was greater than 98.0%, according to the HPLC data [64]. The EchA sample contained no more than 2% natural impurities and other spinochromes in the sea urchin [65]. Here, we used Histochrome^®^ containing 1% echinochrome A in a 0.9% isotonic solution (sodium carbonate, sodium chloride) in a 5 mL vial (10 mg/mL) provided by the G.B. Elyakov Pacific Institute of Bioorganic Chemistry FEB RAS, Vladivostok, Russia. Echinochorome A has been used in various doses (1 or 3mg/kg) for various diseases in previous papers [66]. Among these doses, 3 mg/kg, which is a dose that showed a significant protective effect against ischemic damage of the cardiovascular system in both experimental animals and clinical trials [66], was used in this experiment. Histochrome 0.3 mL/kg/day was diluted in a 0.9% sodium chloride solution to have EchA equivalent of 3 mg/kg/day.

### 4.2. Animal Experiments

Animal experimental procedures were approved by the Inje Medical University Animal Care and Use Committee (approval no. 2021-026). Seven-week-old male diabetic db/db mice (BKS.Cg-Dock7m+/+ Leprdb/J) were used as experimental mice, and seven-week-old male non-diabetic db/m mice (C57BLKS/J) were used as normal controls. Wild-type (WT) mice were purchased from Charles River (Kanagawa, Japan). Mice were randomly divided into three groups: WT, db/db, and db/db + EchA (*n* = 6 mice/group). All animals were housed in a specific pathogen-free facility with controlled temperature (20–24 °C) and humidity (40–70%) on a 12 h light cycle and access to standard laboratory chow and tap water ad libitum. After adaptive feeding for three days, db/db mice with similar blood glucose levels and body weights were randomized into the following groups: untreated db/db control and db/db + EchA. Mice in the db/db + EchA group were administered Histochrome (0.3 mL/kg/day; EchA equivalent of 3 mg/kg/day) daily via intraperitoneal injection for 12 weeks, while WT and db/db control mice were injected with equal amounts of 0.9% sodium chloride solution. The body weight of the mice was measured weekly. At the end of treatment, overnight-fasted mice were sacrificed via exsanguination under anesthesia by inhaling 2% isoflurane in room air. Blood samples and kidney tissues were collected for further experiments, and the kidney weights and tibia lengths were measured.

### 4.3. Fasting Blood Glucose and Glucose Tolerance Test

Every two weeks during treatment, fasting blood glucose levels were measured in all mice after fasting for 16 h (overnight fast). The glucose tolerance test was conducted in the last week of treatment, as previously described [67]. Briefly, 2 μL of blood from the tail vein was collected via tail tipping, and blood glucose levels were measured directly with test strips using a glucose meter (ACCU-CHEK; Roche, Bella Vista, Australia). For the intraperitoneal glucose tolerance test (IPGTT), mice were fasted for 16 h prior to an intraperitoneal injection of a 20% d-glucose solution. Blood glucose levels were determined before the injection (time 0) and at different time points (30, 60, 90, 120, and 150 min).

### 4.4. Measurement of Blood Creatinine, BUN, and Insulin Levels

At the end of the study, the mice were anesthetized via inhalation with 5% isoflurane. Fasting blood specimens were collected from the heart in commercial tubes without anticoagulants, and serum was collected via centrifugation for 10 min at 3000 rpm. The serum was stored at −20 °C until analysis. Finally, serum samples were sent to the Seegene Medical Foundation in Korea to analyze insulin, BUN, and blood creatinine levels.

### 4.5. Histological Analyses

Tissues from the right kidney were excised and washed with phosphate-buffered saline (PBS) to remove the blood, fixed in 4% paraformaldehyde (40 °C, 24 h), and then incubated in 50% ethanol at room temperature. The samples were embedded in paraffin and sectioned into 4 µm thick slices. Kidney sections were stained with hematoxylin and eosin (HE) and Masson’s trichrome (MT). Sections were photographed using a NanoZoomer Digital Slide Scanner (Hamamatsu, Japan). Each section and the relative fibrosis area (% of the total area) were analyzed using image analysis software (ImageJ 1.48 software, NIH, Bethesda, MD, USA).

### 4.6. Measurement of Kidney ATP Levels

An ATP assay kit (ab83355, Abcam) was used to determine the level of ATP in the kidneys, according to the manufacturer’s protocol. Renal ATP levels were normalized to the total protein concentration, and results were presented as nmol/mg total protein.

### 4.7. Measurement of Kidney Malondialdehyde (MDA) and Lipid Hydroperoxide (LPO) Levels

Lipid peroxidation and MDA levels in the kidneys were determined using an MDA assay kit (colorimetric, ab233471, Abcam, Waltham, Boston, USA). An LPO assay kit (Item No. 705003, Cayman, USA) was used to measure the level of LPO in the kidneys. The assays were performed according to their respective manufacturer’s protocols. MDA and LPO levels were normalized to the total protein concentration, and results were presented as μmol and nmol per milligram of protein, respectively.

### 4.8. Western Blotting Analysis

Total protein from the renal tissues of WT and db/db mice was extracted using a RIPA lysis buffer with a protease and phosphatase inhibitor cocktail (Thermo Fisher, Waltham, MA, USA). The extracted proteins were separated via sodium dodecyl sulfate–polyacrylamide gel electrophoresis and transferred to polyvinylidene fluoride membranes. After blocking with 5% skim milk in Tris-buffered saline (TBS) with 0.1% Tween 20, the membranes were incubated with the primary antibodies overnight at 4 °C. The primary antibodies were purchased from Cell Signaling (α-SMA, PGC1a, NRF2, AMPK, p-AMPK Thr172, HO-1, SOD1, SOD2, p-p38 MAPK Thr180/Tyr182, p38MAPK, p-p53 Ser15, p53, p-c-Jun Ser63, c-Jun, and β-actin), Abcam (NOX2, NOX4), and Invitrogen (TGF-β1, PKCι, Col3, Smad2, and p-Smad2 Ser465/467). After washing, the membranes were incubated with the corresponding HRP-conjugated secondary antibodies (Jackson Immuno Research, West Grove, PA, USA), diluted in 5% skim milk, and incubated for 1 h at room temperature. Finally, membranes were washed with TBS containing 0.1% Tween 20. Immuno-detection was performed using an enhanced luminol-based chemiluminescent substrate (WESTSAVE Up, AbFrontier, Seoul, Republic of Korea) and observed using a Bio-Imaging Analyzer System (iBright1500, Thermo-Fisher). β-actin was used as the loading control. Quantification of each band was performed using ImageJ software.

### 4.9. Quantitative Real-time PCR (RT-PCR)

Total RNA from the renal tissues of WT and db/db mice was extracted using TRIzol reagent (Invitrogen, Carlsbad, CA, USA). Total RNA (1.5 µg) was reverse-transcribed to cDNA using a Revert-Aid First Strand cDNA Synthesis kit (AccuPower^®^ RT PreMix, Bioneer, Republic of Korea) following the manufacturer’s protocol. Primers were obtained from Cosmo Genetech (Seoul, Republic of Korea). Their sequences are presented in Table 1. AccuPower^®^ 2X GreenStarTM qPCR Master Mix (Bioneer, Daejeon, Republic of Korea) was used to perform real-time PCR with reactions prepared according to the manufacturer’s protocol. All reactions were performed in triplicates. The cDNA was amplified in 45 cycles using the following settings: 15 s at 95 °C, 30 s at 57 °C, and 30 s at 72 °C. Analysis was performed using CFX Manager™ software (Bio-Rad, Hercules, CA, USA) and Microsoft Excel. Relative RNA levels were normalized to those of β-actin.

### 4.10. Statistical Analysis

Statistical analyses were performed using GraphPad Prism 8.0.1 software (San Diego, CA, USA). All data are expressed as the mean ± standard error of the mean (SEM). Unpaired Student’s *t*-test was used for comparisons between two groups, and one- or two-way analysis of variance (ANOVA) and post hoc Tukey’s test were used for multiple comparisons. Statistical significance was set at *p* < 0.05.

## 5. Conclusions

This study provides evidence suggesting that EchA protects against DN in a db/db mouse model by attenuating renal fibrosis, mitochondrial dysfunction, and oxidative stress mediated by PKCι inhibition, which subsequently suppresses p38 MAPK, the NO4 signaling pathway, and AMPKα activation, which enhances the PGC1α/NRF2/HO-1 pathway (Figure 6). The potential reno-protective mechanism of EchA in a mouse model of type 2 diabetes mellitus may provide a new therapeutic strategy for DN.

**Figure 6 marinedrugs-21-00222-f006:**
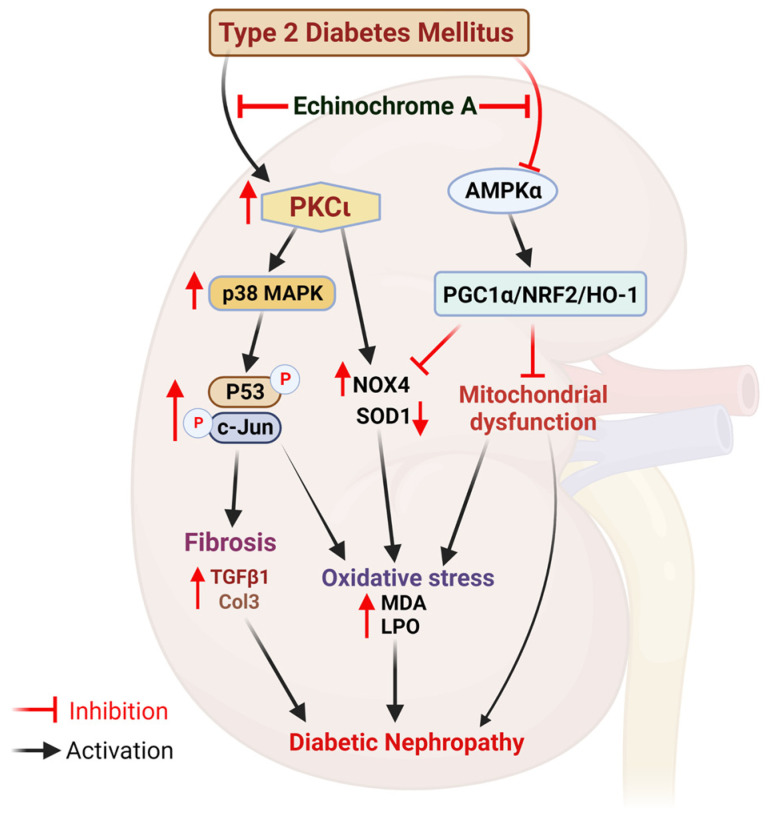
Schematic diagram depicting the potential renal-protective mechanism of EchA in diabetic nephropathy.

## Figures and Tables

**Figure 1 marinedrugs-21-00222-f001:**
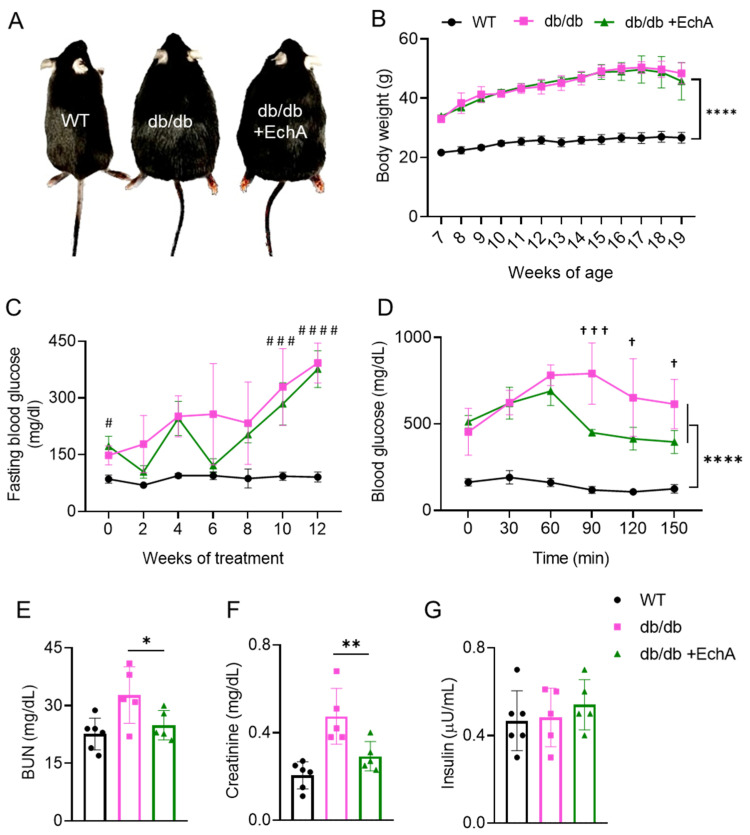
Effect of EchA on body weight, blood glucose, blood urea nitrogen (BUN), creatinine, and insulin levels in db/db mice. (**A**) Morphology of the WT, db/db, and db/db + EchA mice at 19 weeks old and after 12 weeks of treatment. (**B**) Body weight. (**C**) Fasting blood glucose. (**D**) BUN levels. (**E**) Serum creatinine levels. (**F**) Glucose tolerance. (**G**) Insulin levels. Data are presented as mean ± SEM; * *p* < 0.05, ** *p* < 0.01, **** *p* < 0.0001; ^#^
*p* < 0.05, ^###^
*p* < 0.001, ^####^
*p* < 0.0001 in the WT vs. the db/db and db/db + EchA groups. ^ϯ^
*p* < 0.05, ^ϯϯϯ^
*p* < 0.001 in the db/db vs. the db/db + EchA groups (*n* = 6/group).

**Figure 2 marinedrugs-21-00222-f002:**
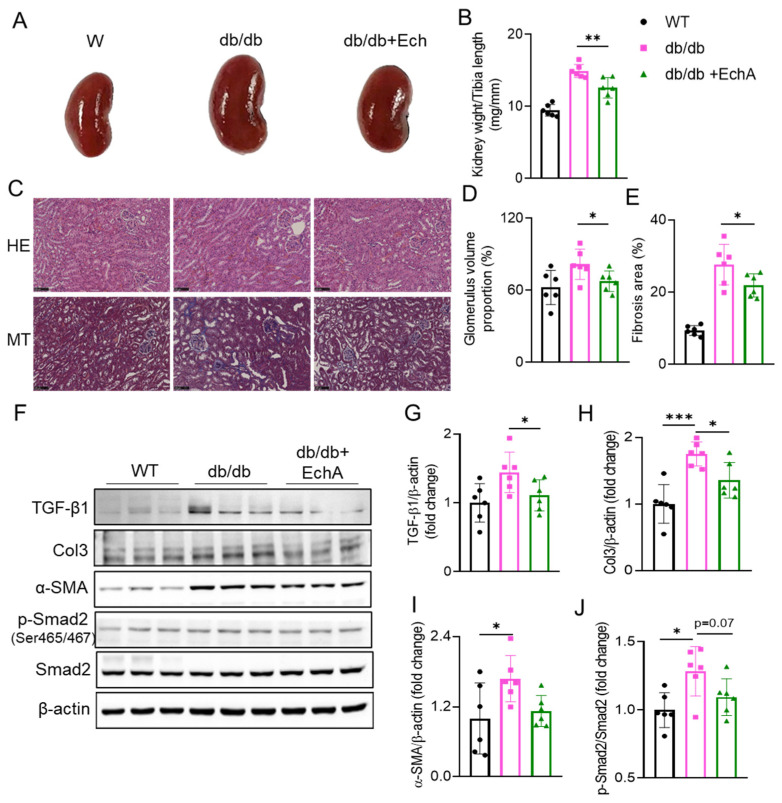
Effects of EchA treatment on renal hypertrophy and fibrosis in db/db mice. (**A**) Representative images of whole kidneys from mice in the different treatment groups. (**B**) The ratio of kidney weight to tibia length of the mice. (**C**) Representative images of HE- and MT-stained kidney sections. Magnification, 400×. The scale bar represents 100 μm. (**D**) Glomerulus volume proportion based on HE staining. (**E**) Renal interstitial fibrosis area based on MT staining. (**F**) Immunoblot analysis of protein expression related to renal fibrosis. (**G**–**J**) Quantitative analysis of transforming growth factor-beta 1 (TGF-β1), collagen III (Col3), alpha smooth muscle actin (αSMA), phospho-suppressor of mothers against decapentaplegic 2 (p-SMAD2), total SMAD2, and β-actin in the kidneys of mice, as determined via Western blotting. Data are presented as mean ± SEM. * *p* < 0.05, ** *p* < 0.01, *** *p* < 0.001 (*n* = 6/group).

**Figure 3 marinedrugs-21-00222-f003:**
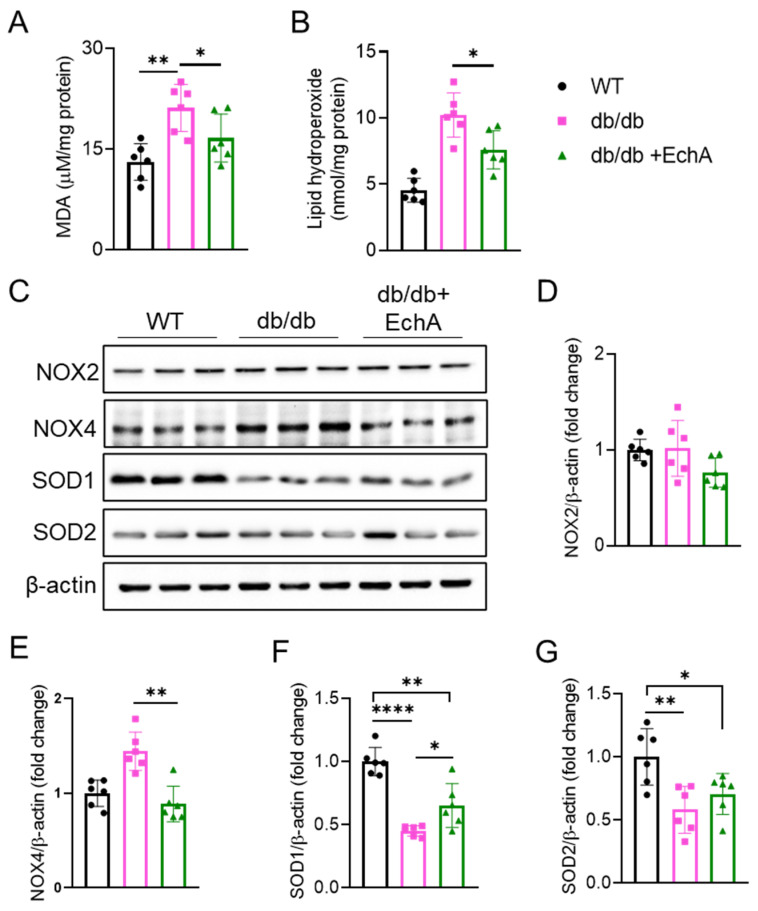
Effects of EchA treatment on oxidative stress in the renal tissue of db/db mice. (**A**) Renal malondialdehyde (MDA) levels. (**B**) Renal lipid hydroperoxide levels. (**C**) Immunoblot analysis of the expression of proteins related to oxidative stress in renal tissues. (**D**–**G**) Quantitative analysis of NADPH oxidase 2 (NOX2), NOX4, superoxide dismutase 1 (SOD1), SOD2, and β-actin protein expression levels in the kidneys of mice, as determined using Western blotting. Data are presented as mean ± SEM. * *p* < 0.05, ** *p* < 0.01, **** *p* < 0.0001 (*n* = 6/group).

**Figure 4 marinedrugs-21-00222-f004:**
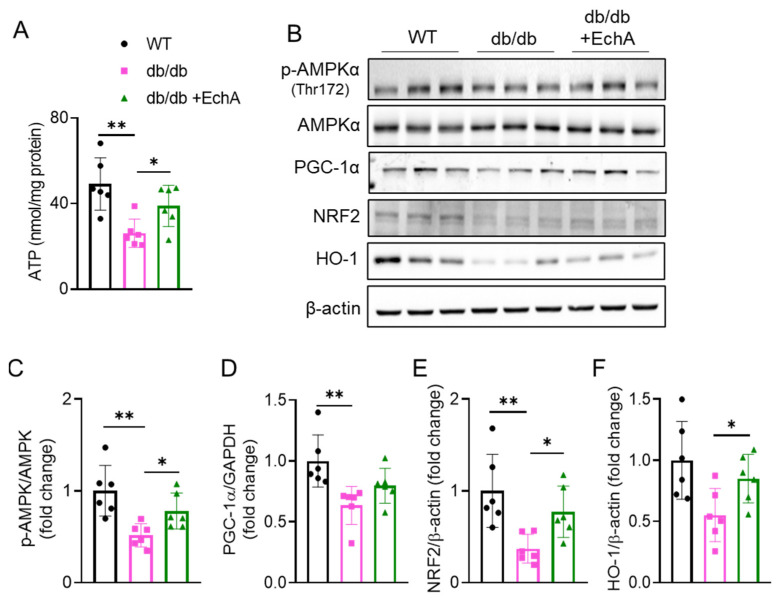
EchA regulates high-glucose-induced p-AMPK, PGC-1α, and NRF2/HO-1 expression and activates ATP production in diabetic kidneys. (**A**) Renal ATP levels. (**B**) Immunoblot analysis of the protein expression of AMP-activated protein kinase (AMPK) in the total and phosphorylated form, peroxisome proliferator-activated receptor gamma coactivator 1-alpha (PGC-1α), nuclear factor erythroid-2-related factor 2 (NRF2), heme oxygenase-1 (HO-1), and β-actin. (**C**–**F**) Statistical analysis of the expression of these proteins mentioned in (**B**). Data are presented as mean ± SEM. * *p* < 0.05, ** *p* < 0.01 (*n* = 6/group).

**Figure 5 marinedrugs-21-00222-f005:**
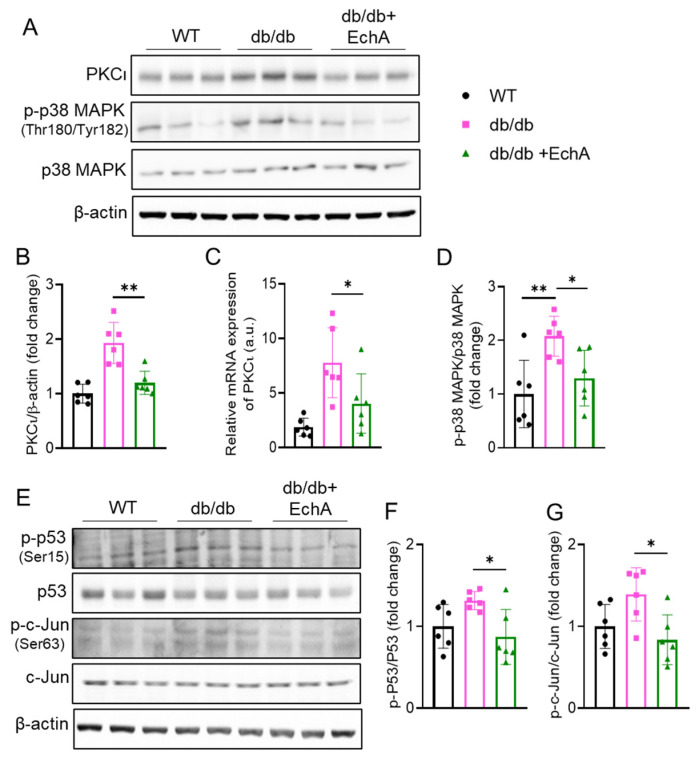
Effect of EchA treatment on PKCι expression and the phosphorylation p53 and c-Jun in the kidney tissues of db/db mice. (**A**) Analysis of the protein expression of PKCι, p-p38 MAPK, p38 MAPK, and β-actin using Western blot analysis. (**B**) Quantitative analysis of PKCι protein expression. (**C**) mRNA expression of PKCι in the kidney as determined using real-time PCR. (**D**) Quantitative analysis of p-p38 MAPK phosphorylation. (**E**) Immunoblot analysis of the protein expression of p-p53, p53, p-c-Jun, c-Jun, and β-actin. (**F**,**G**) Statistical analysis of the expression of the examined proteins. Data are presented as mean ± SEM. * *p* < 0.05, ** *p* < 0.01 (*n* = 6/group).

**Table 1 marinedrugs-21-00222-t001:** Primers used for transcript quantification via RT-PCR.

Gene	Primer Sequences (5′-3′)	
	Forward	Reverse
PKCι	GTTTGAGCAGGCGTCCAATCAC	CAGGAAGTTTTCTCTGTCGCTGC
β-Actin	CATTGCTGACAGGATGCAGAAGG	TGCTGGAAGGTGGACAGTGAGG

## Data Availability

Not applicable.

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
