# Peer review of "Echinochrome A Prevents Diabetic Nephropathy by Inhibiting the PKC-Iota Pathway and Enhancing Renal Mitochondrial Function in db/db Mice"

_marinedrugs, 2023, doi:10.3390/md21040222_

Round 1
Reviewer 1 Report
The study entitled ‘Echinochrome A prevents diabetic nephropathy via inhibiting the PKCι/p38 MAPK and enhancing AMPKα/NRF2/HO-1 pathways in db/db mice’ is interesting and has the ability to discover a novel drug for diabetic nephropathy. Although the experimental design is sound, there are several concerns about the current form of this manuscript.
1. The writing in most parts of the manuscript needs to be improved. There are several typos as well and those need to be corrected.
2. Any abbreviation like BUN needs to be elaborated at its 1st mention.
3. How did the authors select the dose of EchA, 3 mg/kg/day? They need to provide the rationale behind dose selection.
4. In figure 1, panels B, D, E and F should contain the color code indicator for the groups.
5. Is there any role of EchA in the WT mouse? It is very important to include another group of mice, WT+EchA, to determine the role of EchA in the WT mouse.
6. Aldehyde dehydrogenase, ALDH2 detoxifies reactive aldehydes including the 4-hydroxy-2-nonenal (4HNE) that is elevated in diabetes-induced oxidative stress. Like the MDA, 4HNE is also an important biomarker of oxidative stress in diabetes. Therefore, the authors can strengthen their findings by determining the activity of ALDH2 as well as the protein levels of ALDH2 and 4HNE protein adducts in the kidneys of WT, WT+EchA, db/db, and db/db+EchA mice.
Reviewer 2 Report
I have read the manuscript and have a few questions and comments.
1. The title of the manuscript is hard to read. Please don't use too many abbreviations.
2. Section 4 does not contain a materials section. Please complete the "Materials" section to include information about the test substance, including date of extraction/synthesis, literature reference, expiration date, storage conditions, purity, and presence of impurities.
3. In section 4.1, please include information about the number of animals in the group. How the test compound was administered to animals, in what solvent.
4. The article must be based on careful and extensive research using proper controls.
Please complete the study with an additional positive control group. Compare the control group data with your data.
5. The scientific value of single dose studies is very low. Please complete the study with different dose groups. Compare the control group data with your data.
6. Specify brands of all used equipment, manufacturer, country.
7. How can you explain the variation in glucose in the db/db+Ech A group in 4 weeks of treatments? At what time was the blood taken from the animals? Was it on an empty stomach or after a meal?
8.Ech A is registered as a medicinal product and is approved for medicinal use in Russia. Please include information in the introduction to increase interest in this substance of animal origin.
Round 2
Reviewer 1 Report
Thanks, to the authors for addressing all the concerns I had about the 1st version of the manuscript. Therefore, the manuscript is suitable to be published in its current form.
Author Response
We appreciate the positive feedback from the reviewer. Thank you very much!
Reviewer 2 Report
I have read the revised version of the manuscript. The authors did not fully answer my questions.
1. Animal experiment performed using a single dose. The significance of such data is insignificant.
Please provide the multi-dose experiment.
2. The experiment was performed without using the proper control group. Please experiment with the control group. Use a commonly used diabetes drug as a positive control.
Compare the efficacy of your drug with the efficacy of the proper control group.
3. In section 4.1, the authors must clearly indicate whether they used the Histochrome drug or the echinochrome A substance as the object of the study.
